



# Constraining the Twomey effect from satellite observations: Issues and perspectives

Johannes Quaas[1], Antti Arola[2], Brian Cairns[3], Matthew Christensen[4], Hartwig Deneke[5],
Annica M. L. Ekman[6], Graham Feingold[7], Ann Fridlind[3], Edward Gryspeerdt[8], Otto Hasekamp[9],
Zhanqing Li[10], Antti Lipponen[2], Po-Lun Ma[11], Johannes Mülmenstädt[11], Athanasios Nenes[12,13],
Joyce Penner[14], Daniel Rosenfeld[15], Roland Schrödner[5], Kenneth Sinclair[3,16], Odran Sourdeval[17],
Philip Stier[4], Matthias Tesche[1], Bastiaan van Diedenhoven[3], and Manfred Wendisch[1]

[1]Universität Leipzig, Leipzig Institute for Meteorology
[2]Finnish Meteorological Institute
[3]NASA Goddard Institute for Space Studies, New York
[4]University of Oxford
[5]Leibniz Institute for Tropospheric Research, Leipzig
[6]Stockholm University, Department of Meteorology and Bolin Centre for Climate Research
[7]NOAA Boulder
[8]Space and Atmospheric Physics Group, Imperial College London
[9]SRON Netherlands Institute for Space Research, Utrecht
[10]University of Maryland, College Park
[11]Pacific Northwest National Laboratory, Richland
[12]School of Architeture, Civil & Environmental Engineering, Ecole Polytechnique Fédérale de Lausanne, Switzerland
[13]Institute of Chemical Engineering Sciences, Foundation for Research and Technology Hellas, Patras, Greece
[14]University of Michigan, Ann Arbor
[15]Hebrew University of Jerusalem
[16]Universities Space Research Association (USRA), Columbia, MD 21046, USA
[17]Université de Lille, CNRS, UMR 8518 - LOA - Laboratoire d'Optique Atmosphérique, Lille, France

**Correspondence:** Johannes Quaas (johannes.quaas@uni-leipzig.de)

**Abstract.**

The Twomey effect describes the radiative forcing associated with a change in cloud albedo due to an increase in anthropogenic aerosol emissions. It is driven by the perturbation in cloud droplet number concentration ($\Delta N_{d,ant}$) in liquid-water clouds and is currently understood to exert a cooling effect on climate. The Twomey effect is the key driver in the effective

radiative forcing due to aerosol-cloud interactions which also comprises rapid adjustments. These adjustments are essentially the responses of cloud fraction and liquid water path to $\Delta N_{d,ant}$ and thus scale approximately with it. While the fundamental physics of the influence of added aerosol particles on the droplet concentration ($N_d$) is well described by established theory at the particle scale (micrometres), how this relationship is expressed at the large scale (hundreds of kilometres) $\Delta N_{d,ant}$ remains uncertain. The discrepancy between process understanding at particle scale and insufficient quantification at the climate-

relevant large scale is caused by co-variability of aerosol particles and vertical wind and by droplet sink processes. These operate at scales on the order of 10s of metres at which only localized observations are available and at which no approach exists yet to quantify the anthropogenic perturbation. Different atmospheric models suggest diverse magnitudes of the Twomey





effect even when applying the same anthropogenic aerosol emission perturbation. Thus, observational data are needed to quantify and constrain the Twomey effect. At the global scale, this means satellite data. There are three key uncertainties in

determining $\Delta N_{d,ant}$, namely the quantification (i) of the cloud-active aerosol – the cloud condensation nuclei concentrations (CCN) at or above cloud base –, (ii) of $N_d$, as well as (iii) the statistical approach for inferring the sensitivity of $N_d$ to aerosol particles from the satellite data. A fourth uncertainty, the anthropogenic perturbation to CCN concentrations, is also not easily accessible from observational data. This review discusses deficiencies of current approaches for the different aspects of the problem and proposes several ways forward: In terms of CCN, retrievals of optical quantities such as aerosol optical depth

suffer from a lack of vertical resolution, size and hygroscopicity information, the non-direct relation to the concentration of aerosols, the impossibility to quantify it within or below clouds, and the problem of insufficient sensitivity at low concentrations, in addition to retrieval errors. A future path forward can include utilizing colocated polarimeter and lidar instruments, ideally including high spectral resolution lidar capability at two wavelengths to maximize vertically resolved size distribution information content. In terms of $N_d$, a key problem is the lack of operational retrievals of this quantity, and the inaccuracy of

the retrieval especially in broken-cloud regimes. As for the $N_d$ - to - CCN sensitivity, key issues are the updraught distributions and the role of $N_d$ sink processes, for which empirical assessments for specific cloud regimes are currently the best solutions. These considerations point to the conclusion that past studies using existing approaches have likely underestimated the true sensitivity and, thus, the radiative forcing due to the Twomey effect.

## 1    Introduction

Cloud droplets in liquid-water clouds form on cloud condensation nuclei (Aitken, 1880), a subset of the atmospheric aerosol particle population. The formation of cloud droplets in thermodynamic equilibrium is established textbook knowledge (Köhler, 1936). Whether an aerosol particle acts as a cloud condensation nucleus (CCN) at a given supersaturation depends on its size and chemical composition which determines the particle hygroscopicity (Dusek et al., 2006; Ma et al., 2013). If CCN concentrations at one supersaturation level are known, CCN concentrations at other supersaturation levels approximately scale with

it (Twomey, 1959) if the CCN distribution can be approximated by one log-normal mode. Here, we implicitly consider a supersaturation level of 0.2% unless otherwise stated. Supersaturation is generated in the large majority of clouds by updraughts. The rare exceptions are formation due to radiative cooling (mainly fog events) or the mixing of cold and dry with warm and moist air masses. Cloud-scale updraughts originate in most cases from turbulence, convection, or gravity waves. Vertical wind, $w$, exhibits a large heterogeneity across temporal and spatial scales (Tonttila et al., 2011; Moeng and Arakawa, 2012). For

a given probability density function (PDF) of updraughts, in an adiabatic air parcel with no active collision-coalescence, the addition of extra CCN will generally lead to a monotonic increase in cloud droplet number concentration, $N_d$ (Twomey and Warner, 1967). The approximate functional form of the dependence of $N_d$ on CCN concentration is then logarithmic, since the increase in $N_d$ associated with activation of additional aerosol leads to a depletion of the maximum supersaturation (Twomey, 1959).





The CCN concentration in the atmosphere is increased by anthropogenic emission of aerosols and aerosol precursor gases (Boucher et al., 2013). This leads to enhanced $N_d$, unless aerosol particle concentrations are high and updraughts weak (Ghan et al., 1998; Feingold et al., 2001; Reutter et al., 2009). In turn, cloud albedo ($\alpha_c$, the fraction of solar radiative energy reflected back to space by clouds in relation to that incident at the cloud top) increases, as it is a monotonically increasing function of $N_d$. Following Platnick and Twomey (1994) and Ackerman et al. (2000),

$$\frac{\partial \ln \alpha_c}{\partial \ln N_d} = \frac{1}{3}\left(1 - \alpha_c\right), \tag{1}$$

a formulation which relies (i) on a two-stream radiative transfer approximation, and (ii) the assumption that clouds obey vertical stratification that scales with an adiabatic one and that is horizontally homogeneous. Eq. 1 is expressed as a partial derivative: other cloud quantities – notably cloud horizontal extent and cloud water path – are considered constant.

These two facts – $N_d$ is a monotonic function of CCN and $\alpha_c$ is a monotonic function of $N_d$ – imply that the anthropogenic increase in CCN concentrations causes a negative (cooling) radiative forcing due to aerosol-cloud interactions, RF$_{aci}$ (Boucher et al., 2013), denoted as $\mathcal{F}_{aci}$ (Bellouin et al., 2020b). It can be approximately (neglecting absorption in the column above the cloud after scattering at cloud top) written as (Quaas et al., 2008; Bellouin et al., 2020b):

$$\mathcal{F}_{aci} = F_s^{\downarrow} \cdot \frac{\partial \alpha_c}{\partial \ln N_d} \cdot \frac{\partial \ln N_d}{\partial \ln a} \cdot \Delta \ln a_{ant} \tag{2}$$

with the downward solar radiative flux density (irradiance) above clouds, $F_s^{\downarrow}$, and a quantitative description of CCN denoted here as $a$. The relative anthropogenic perturbation to $a$ is denoted $\Delta \ln a_{ant}$. This formulation assumes (i) that only the solar spectrum is relevant, which is well justified for the optically thick, liquid water clouds considered here, since an $N_d$ perturbation only marginally changes the cloud radiative effect in the terrestrial spectrum of an optically thick cloud; and (ii) that there is one liquid water cloud layer that determines the effect so that the problem can be considered as purely horizontal in space. In contrast to the formulation by Bellouin et al. (2020b), we consider the problem as horizontally variable in space $(x, y)$ and in time $(t)$, i.e. $\mathcal{F}_{aci} = \mathcal{F}_{aci}(x, y, t)$. If Eq. 2 is assessed from temporally-sparse satellite data, a proper integration over temporally varying zenith angles and cloud diurnal cycles is necessary.

RF$_{aci}$ is often referred to as "Twomey effect" (Twomey, 1974) and also called "(first) aerosol indirect effect" or "cloud albedo effect" (Lohmann and Feichter, 2001). Atmospheric models simulate a large range for RF$_{aci}$ (Gryspeerdt et al., 2020; Smith et al., 2020). It is, thus, necessary to constrain the Twomey effect quantitatively based on observations. Only satellites can provide global observational data that could be used to quantify the global RF$_{aci}$ (Stephens et al., 2019).

The Twomey effect has been assessed in many studies (starting with Bréon et al., 2002) in terms of cloud droplet effective radius, $r_e$, rather than using $N_d$. This is plausible as, for idealized vertical profiles of droplet size distributions (e.g., vertically constant or adiabatically increasing profiles), cloud optical depth and cloud albedo are easily expressed in terms of $r_e$ (Hansen and Travis, 1974; Stephens, 1978). Given that $r_e$ is closely related to light-scattering properties of clouds, this quantity is operationally retrieved from remote-sensing observations (Nakajima and King, 1990). However, $r_e$ is not just a function of $N_d$ but also varies with cloud liquid water path, $L$ (Brenguier et al., 2000). It is thus necessary to formulate the problem for constant $L$, which is difficult to realize in data analysis from observations that are limited in time and space, or to selected





cloud scenarios, so that datasets stratified by $L$ become too small for meaningful analysis (Quaas et al., 2006; McComiskey and Feingold, 2012; Liu and Li, 2019).

Among the four factors on the right-hand side of Eq. 2, the first one, $F_s^{\downarrow}$, is well quantified for each given latitude, longitude, and time. The second one, $\partial \alpha_c / \partial \ln N_d$, can be evaluated using Eq. 1 (Bellouin et al., 2020b; Hasekamp et al., 2019a), or alternatively by radiative-transfer simulations (Mülmenstädt et al., 2019). This implies that the two key problems in determining $RF_{aci}$ are the quantification of the anthropogenic perturbation of CCN, $\Delta \ln a_{ant}$, and the sensitivity of $N_d$ to CCN perturbations, $\beta = \partial N_d / \partial \ln a$ (Feingold et al., 2001). Taken together, this is the distribution of the anthropogenic perturbation of $N_d$

(here expressed in absolute, not relative terms):

$$\Delta N_{d,ant} = \left. \frac{\partial N_d}{\partial \ln a} \right|_w \cdot \Delta \ln a_{ant} = \beta(w) \cdot \Delta \ln a_{ant}. \tag{3}$$

The plausible range of the sensitivity is $0 \leq \beta \leq 1$, except for heavily polluted situations (where it may become negative; Feingold et al., 2001), or when giant CCN play an important role (Ghan et al., 1998; Betancourt and Nenes, 2014; Gryspeerdt et al., 2016; McCoy et al., 2017) where competition for water vapor during droplet formation is at its strongest. Under such

conditions, even models and parameterizations of the process are challenged the most (Betancourt and Nenes, 2014).

The aerosol forcing has to be evaluated at a scale much larger than an individual cloud. One of the key reasons for this is that there is currently no way to use satellite data to determine the anthropogenic fraction of the CCN population for a single air parcel. Methods applying model information, or data-tied approaches such as Bellouin et al. (2013) instead use the scale of model resolution or aggregate data resolution which is typically of the order of $1° \times 1°$ (or about $100 \times 100 \, \text{km}^2$). The problem

formulated in Eq. 3 then has to be reformulated, using an overbar to denote the averaging over a $1° \times 1°$ grid-box as

$$\overline{\Delta N_{d,ant}} = \overline{\left[ \int_{w=-\infty}^{\infty} \left. \frac{\partial N_d}{\partial \ln a} \right|_w \mathcal{P}(w)\,\mathcal{P}(a)\,\mathrm{d}w \right] \Delta \ln a_{ant}} = \overline{\beta} \cdot \overline{\Delta \ln a_{ant}} \tag{4}$$

which considers the mean sensitivity of $N_d$ to CCN, $\overline{\beta}$, given the probability density function (PDF) of cloud-base vertical wind, $w$ in the grid-box, $\mathcal{P}(w)$, the PDF of CCN at cloud base within the scene, $\mathcal{P}(a)$, and the anthropogenic perturbation of the CCN concentration at the grid-box scale, $\overline{\Delta \ln a_{ant}}$. Note in the above equation, $\beta$ is assumed independent of $\ln a_{ant}$,

which assumes that $\mathcal{P}(w)$ is independent of cloud properties (primarily, liquid water content), which applies to stratus clouds (Morales and Nenes, 2010) but not in general. Similarly, the covariance of $\mathcal{P}(w)$ and $\mathcal{P}(a)$ may not be zero (e.g., Kacarab et al., 2020 - in addition to Bougiatioti et al., 2020). All the above suggests that observation of $\beta$ at a cloud parcel scale is not directly transferrable to the large-scale for an assessment of the Twomey effect. Rather, $\overline{\beta}$ has to be estimated.

Beyond $RF_{aci}$, aerosol-cloud interactions also lead to rapid adjustments: once cloud droplet size distributions are altered due

to anthropogenic CCN, cloud microphysical and dynamical processes are modified as well (Albrecht, 1989; Ackerman et al., 2000; Heyn et al., 2017; Mülmenstädt and Feingold, 2018). Aerosols can induce transitions between cloud regimes, for instance by changing drizzle behavior (Rosenfeld et al., 2006; Feingold et al., 2010; Wood et al., 2011). The direction and magnitude of these changes depends on the cloud state and regime, because responses to aerosol changes occur due to processes spanning a range from microphysics to the mesoscale (Christensen and Stephens, 2012; Kazil et al., 2011; Wang et al., 2011). These





processes include precipitation suppression (Albrecht, 1989), rapid feedbacks involving cloud-top entrainment (Ackerman et al., 2004; Bretherton et al., 2007; Hill et al., 2009; Bulatovic et al., 2019), and rapid feedbacks involving cloud lateral entrainment (Xue and Feingold, 2006; Small et al., 2009) as well as responses in dynamics (Xue et al., 2008; Stevens and Feingold, 2009; Wang and Feingold, 2009). If one considers also deep clouds, further intricate cloud adjustments may occur that are not considered here (e.g., Ekman et al., 2011; Fan et al., 2013; Yan et al., 2014). As a result of these adjustment processes,

cloud horizontal extent (Gryspeerdt et al., 2016) and liquid water path (Gryspeerdt et al., 2019) respond to perturbations in $N_{\mathrm{d}}$. The sum of $\mathrm{RF_{aci}}$ and the radiative effects of these adjustments is the effective radiative forcing due to aerosol-cloud interactions, $\mathrm{ERF_{aci}}$ (Boucher et al., 2013). Based on modelling and data analysis, it is evident that the adjustments and, thus, also $\mathrm{ERF_{aci}}$, scale with $\Delta N_{\mathrm{d,ant}}$ (Bellouin et al., 2020b; Gryspeerdt et al., 2020; Mülmenstädt et al., 2019). Analysis of model data shows that the rapid adjustments due to other contributions (small- to mesoscale circulation changes, thermodynamic

changes) are small (Heyn et al., 2017; Mülmenstädt et al., 2019). Even so, thermodynamic and dynamic adjustments to aerosol changes can still have an important impact on droplet formation - especially under conditions where droplet formation is largely velocity-limited (Kacarab et al., 2020; Bougiatioti et al., 2020).

Despite the fact that the activation of an individual CCN to form a droplet is well understood in thermodynamic equilibrium (Köhler, 1936), it is not clear how $N_{\mathrm{d}}$ responds to perturbations of CCN at the scale of a cloudy air parcel, an entire cloud,

or at the scale of a cloud field up to the large scale of order of $1° \times 1°$ as used in Eq. 4. A one-to-one relationship between CCN in the updraught below cumulus and $N_{\mathrm{d}}$ above cloud base within the cumulus has been observed (Werner et al., 2014); although even at the cloud updraft scale this relationship could be a convolution of the effect of CCN on droplet number, vertical velocity variability and lateral entrainment (Morales et al., 2011). At a larger scale, this relation is less pronounced (Boucher and Lohmann, 1995), consistent with the expectation from Eq. 4. In turn, there may be co-variability of updraughts

and aerosol concentrations that lead to larger $\bar{\bar{\beta}}$ compared to situations with constant $w$ (Kacarab et al., 2020; Bougiatioti et al., 2017, 2020).

Ground-based remote sensing methods provide data to infer the sensitivity term $\beta$ from long-term observations (Feingold et al., 2003; McComiskey et al., 2009; Schmidt et al., 2015; Liu and Li, 2018). However, this approach is limited to individual sites and cloud regimes. In consequence, when investigating the global radiative forcing relevant for climate studies, the

sensitivity term necessarily is derived from satellite remote sensing (Nakajima and Schulz, 2009).

This leads to a number of problems and challenges discussed in more detail in the following sections:

- **Retrieval of CCN.** The first issue is the missing coincidence of cloud and aerosol retrievals. Usually, no aerosol is retrieved below or within clouds. It is thus questionable how representative aerosol in cloudless scenes is for (neighboring) cloud-base CCN. The second issue is the imperfect nature of proxies for CCN. Often the aerosol optical depth (AOD,

see below) or a variant thereof is used, which can only imperfectly be related to CCN due to differences in sensitivity and the lack of vertical resolution.

- **Retrieval of $N_{\mathrm{d}}$.** There are (i) retrieval errors and biases in $N_{\mathrm{d}}$, which depend on cloud regimes, and (ii) one needs to consider the link between $N_{\mathrm{d}}$ as formed by CCN activation at cloud base, and the retrieved cloud-top $N_{\mathrm{d}}$. Cloud-top





$N_\mathrm{d}$ ($N_\mathrm{d,top}$) is the one that determines the scattering of sunlight and, thus, is relevant for the top-of-atmosphere cloud

radiative effect. It differs from cloud-base $N_\mathrm{d}$ ($N_\mathrm{d,base}$) in conditions where $N_\mathrm{d}$ sinks such as precipitation or mixing
        play a role. When using $r_\mathrm{e}$ rather than $N_\mathrm{d}$ the additional problem of stratification by retrieved $L$ arises.

  – **Cloud-regime dependence.** Cloud base droplet concentration, $N_\mathrm{d,base}$, is a function of both CCN and updraught, and
    $N_\mathrm{d,top}$ further a function of $N_\mathrm{d}$ sinks such as precipitation formation and entrainment-mixing. Thus, one needs to un-
    derstand how the characteristics of $w$ and its PDF, as well as precipitation and mixing processes depend on cloud regime

and how this may be used for an empirical estimation of $\bar{\beta}$.

  – **Aggregation scale.** The relation of aggregate quantities is not the same as the aggregate relation, and, thus, one needs
    to determine how to derive $\bar{\beta}$ optimally from remote sensing data (Grandey and Stier, 2010; McComiskey and Feingold,
    2012).

        In practical terms, one further needs to assess to which extent a simple scalar sensitivity metric is sufficient, or whether a
joint-PDF approach is preferable (McComiskey and Feingold, 2012; Gryspeerdt et al., 2017).

        Beyond these questions which are discussed in the following sections, it is necessary to quantify the anthropogenic pertur-
bation to CCN, $\Delta \ln a_\mathrm{ant}$, which is not easily quantified from observations. The key problem is that there is little potential to
observe an atmosphere unperturbed by anthropogenic emissions (Carslaw et al., 2013, 2017). Some studies attempt to quan-
tify the anthropogenic perturbation to the column aerosol light extinction, or aerosol optical depth (AOD; $\tau_a$), in a data-tied

approach (Kaufman et al., 2005; Bellouin et al., 2005, 2013; Kinne, 2019). Such approaches rely on simplifying parameter-
isations, such as the assumption that small-mode aerosol particles are predominantly anthropogenic. The other option is to
estimate it from simulations (Quaas et al., 2009b; Gryspeerdt et al., 2017). There are some indirect ways to infer the anthro-
pogenic impacts on $N_\mathrm{d}$ (Quaas, 2015), such as from trends (Krüger and Graßl, 2002; Bennartz et al., 2011) or periodicity in
anthropogenic emissions such as the weekly cycle (Quaas et al., 2009a). Hence, models are involved in determining an an-

thropogenic perturbation of CCN concentrations, which can even be attempted for individual weather events (Schwartz et al.,
2002). In any case, it seems impossible to know the anthropogenic perturbation to the aerosol at the scale of an air parcel; it
rather is possible only at larger, aggregate scales. The remainder of this review will focus on the sensitivity term $\bar{\beta}$.

## 2   Remote sensing of CCN concentrations

The aerosol quantity most accessible to satellite remote sensing is AOD (Kaufman et al., 2002). It is derived from the multi-

spectral reflectance of the Earth-atmosphere system using the incident solar radiation and retrieving or assuming surface albedo
characteristics as well as aerosol absorption coefficient and scattering phase functions. There are four key issues with using the
retrieved AOD for estimating the $N_\mathrm{d}$ to CCN sensitivity, which will be discussed in the following subsections, namely:

  – **AOD is the vertical integral of the extinction coefficient.** For the sensitivity of $N_\mathrm{d}$ to the aerosol, one needs to know
    the vertical distribution of the CCN concentration, most importantly the CCN at cloud base.





– **AOD is an optical integral and does not provide information on the aerosol size distribution and its hygroscopicity.**
        The use of AOD does not isolate aerosol particles that have the size and chemical composition to serve as CCN. It is also
        affected by aerosol swelling due to hygroscopic growth.

        – **AOD can be derived only for pixels determined as cloud-free.** The degree to which this correlates with the CCN at
        the base of (neighbouring) clouds is questionable. In addition, retrieved AOD can show a positive bias due to enhanced
reflectance from neighbouring cloudy pixels or due to the lack of detecting spurious clouds in a retrieval scene.

        – **The optical signal is very weak at low concentrations.** Therefore, retrievals become more and more uncertain below a
        certain aerosol load, especially over land and in situations with variable or uncertain surface albedo.

At aggregate scales, i.e. for monthly averages over regions, AOD from ground-based remote sensing retrievals (AERONET;
Holben et al., 2001) correlates well with CCN surface measurements (Andreae, 2009; Shen et al., 2019). Similar results were
also reported for aircraft measurements (Clarke and Kapustin, 2010; Shinozuka et al., 2015). However, at shorter timescales or
less spatial aggregation, there are significant deviations from a perfect correlation (Liu and Li, 2014). AOD due to aerosol light
extinction is determined by the vertical integral of the extinction cross section, proportional to the vertical integral of the second
moment of the aerosol size distribution. In turn, for a given chemical composition of aerosol particles, the CCN concentration
is the zeroth moment of the size distribution for particles exceeding a size threshold that depends on supersaturation. In the
following, the different problems are discussed in more detail, together with options for a better proxy for CCN from satellite
remote sensing.

## 2.1   Vertical co-location

Stier (2016) investigated the correlation between AOD and CCN as represented in a climate model. He confirmed a mostly
positive correlation of the temporal variability of the two quantities, although in some regions the correlation is low or even
negative. A key reason for the partly low correlation is the fact that AOD is a vertically integrated quantity and may include
aerosol layers that are not interacting with clouds. A similar result was reported from a statistical analysis of satellite data:
cloud microphysical parameters correlate well with aerosol properties only if the vertical alignment of the aerosol and cloud
layers is accounted for (Costantino and Bréon, 2010, 2013). Ship measurements of CCN and microwave-retrieved $N_d$ at cloud
base between Los Angeles and Hawaii show weaker $\beta$ metric as the boundary layer deepens thus indicating that surface
aerosol measurements become more inadequate to represent aerosol variability at cloud base as the boundary layer deepens
(Painemal et al., 2017), ot that the updrafts become high enough to activate smaller aerosols than the accumulation mode.
In-situ observations suggests that AOD may even be anticorrelated with CCN at cloud base (Kacarab et al., 2020).

A way forward is the use of spaceborne vertically resolved observations such as lidar measurements (Shinozuka et al.,
2015; Stier, 2016). The Cloud-Aerosol Lidar and Infrared Pathfinder Satellite Observations (CALIPSO; Winker et al., 2009)
lidar retrieves aerosol backscatter profiles, and thus is capable of identifying aerosol layers (Costantino and Bréon, 2010).
Profiles of aerosol particle extinction are inferred from these backscatter profiles by using typical extinction-to-backscatter
ratios based on aerosol type. However, the signal is not sensitive to smaller aerosol concentrations which hampers a quantitative





analysis at large scale (Watson-Parris et al., 2018; Ma et al., 2018). For situations with sufficient aerosol loading for reliable CALIPSO aerosol profile observations, methods for retrieving CCN concentrations from ground-based lidar measurements
can be adapted (Feingold and Grund, 1994; Lv et al., 2018; Haarig et al., 2019). These methods apply empirical extinction-to-particle-concentration relationships to obtain input for CCN concentrations for different aerosol types (Mamouri and Ansmann, 2016). In the future, the EarthCARE satellite mission currently scheduled for launch in 2022 (Illingworth et al., 2015; Hélière et al., 2017) shows promise to extend and improve upon the success of the CALIPSO mission. Its Atmospheric Lidar (ATLID) is a linearly polarized high-spectral resolution lidar (HSRL) operating at a wavelength of 355 nm, allowing to directly infer
profiles of aerosol extinction without use of assumptions, thereby substantially increasing the retrieval accuracy. While a similar sensitivity to aerosol load is expected for ATLID and CALIOP observations during nighttime, ATLID promises a better daytime sensitivity. EarthCARE is also expected to provide better distinction between optically thin clouds and aerosols than CALIPSO (Reverdy et al., 2015). Airborne measurements have shown that further utilizing HSRL at more than one wavelength (extending beyond ATLID) would provide substantial additional information content for retrieving vertically resolved aerosol parameters,
especially when combined with polarimeter measurements (Burton et al., 2016). From the passive-remote sensing perspective, promising results have been obtained for retrievals of aerosol vertical information from near-ultra-violet polarimetry (Wu et al., 2016), although the quality degrades for small aerosol concentrations. Passive observations with high spectral resolution within the oxygen A absorption band around 760 nm can also be used to infer aerosol layer height (Hollstein and Fischer, 2014; Geddes and Bösch, 2015). In particular, an operational aerosol layer height product is now available from the Tropospheric Monitoring
Instrument (TROPOMI) flown on the Sentinel-5p mission (Sanders et al., 2015). Also, a recent study presents promising results based on Orbiting Carbon Observatory 2 (OCO-2) observations (Zeng et al., 2020). In particular, a combination of such approaches, e.g. passive polarimetry and active lidar observations (Stamnes et al., 2018) or multi-angle polarimetry and oxygen A band observations as planned for NASA's Plankton, Aerosol, Cloud, ocean Ecosystem (PACE) mission (Remer et al., 2019) shows promising potential. Retrievals could also combine observations and model adjoints to constrain below-cloud aerosol
number, which is directly relevant for aerosol-cloud interactions (Saide et al., 2012).

In summary, the lack of vertical co-location between retrieved CCN proxy and clouds leads to an underestimate in $N_{\mathrm{d}}$ – CCN sensitivity (Costantino and Bréon, 2010). Model studies suggest that this bias may be approximately cancelled by a corresponding bias in the anthropogenic component of the cloud base CCN (Gryspeerdt et al., 2017). However, the extent of this cancellation in current observational studies is unknown and requires further investigation. For an accurate estimation of $\bar{\beta}$
the use of lidar retrievals seems to be the best way forward, while additional information on the vertical distribution of aerosol can also be gained from present and upcoming passive satellite instruments.

## 2.2  Horizontal co-location

In studies examining $\beta$ from satellite data, spatial aggregates are considered (i.e., $\bar{\beta}$ as in Eq. 4), in which the aerosol retrievals in the cloud-free pixels are averaged at a coarse resolution (such as $1°$) and taken to define the relation with $N_{\mathrm{d}}$ retrievals in
the same grid-box (Quaas et al., 2008). This assumes that the aerosol population is horizontally homogeneous at such large scales. According to Anderson et al. (2003), this is often the case. It has been confirmed from aircraft data for the stratocumulus



cases investigated by Shinozuka et al. (2020). However, CCN is consumed when droplets activate and aerosol is scavenged when clouds precipitate. Hence, the assumption of aerosol concentration horizontal homogeneity is questionable at least in precipitating clouds.

It is the aerosol in air masses before cloud particles form that is relevant to compute the aerosol impact on $N_d$ (Gryspeerdt et al., 2015). In one of the early aerosol-cloud interaction studies from satellite data (Bréon et al., 2002) used trajectories to identify cloudless situations in which aerosol retrievals were possible for air masses that later formed clouds. This is a promising solution but it requires much more effort than the simpler co-location assumptions. It also requires reliable, high-resolution information about atmospheric trajectories. Another complication is that the formation rate of secondary aerosol is

enhanced by aqueous phase reactions, potentially enhancing aerosol concentrations in the vicinity of clouds (Jeong and Li, 2010).

Altogether, the lack of horizontal co-location may imply somewhat too low $\bar{\beta}$ due to the potential de-correlation of CCN concentrations and $N_d$ in situations with spatially heterogeneous aerosol. The consideration of backward trajectory analysis seems the best option to address the issue since there is no solution yet to retrieve aerosols below or within clouds from satellite.

## 2.3    Hygroscopic growth of aerosol particles

The extinction of solar radiation by aerosol particles is a strong function of the hygroscopic growth of the particles. Haze particles attenuate much more sunlight compared to the same aerosol particle ensemble in dry conditions. AOD is thus heavily influenced by the variability of relative humidity. The light extinction caused by dry particles (at relative humidities below 30%) is much better correlated to CCN concentrations than the extinction of particles at ambient relative humidity (Shinozuka

et al., 2015). Liu and Li (2018) showed that using total AOD compared to dry AOD as a CCN proxy when estimating $\bar{\beta}$ from measurements at different Atmospheric Radiation Measurements (ARM) sites resulted in a 23% underestimate. A way forward is to apply parameterisations in terms of retrievals of relative humidity to account for the aerosol swelling. These, however, need information about aerosol hygroscopicity and relative humidity at the appropriate scale. Another alternative would be to retrieve the amount of aerosol water, making use of the real part of the refractive index (Schuster et al., 2009). This

would allow to translate the size distribution of humidified aerosol particles to the corresponding dry size distribution. In the near future, accurate refractive index retrievals are expected from polarimeters such as the SPEXone instrument on the NASA PACE mission (Hasekamp et al., 2019b; Werdell et al., 2019), to be launched in 2022.

Summarizing, using AOD as a proxy for CCN results in low-biased estimates of $\bar{\beta}$ due to aerosol swelling. Approaches to parameterise the dry aerosol properties on the basis of the humidified one can help alleviate the problem.

## 2.4    Approaches using aerosol index, column-CCN, reanalysis or cloud-base updraught

The aerosol index (AI[1]) is defined as the product of AOD and the Ångström exponent (Deuzé et al., 2001). This latter quantity is the slope of the spectral variation in AOD and is typically larger for smaller particles (Ångström, 1929). AI is more weighted

---

[1]The difference in the measured radiance in the ultra-violet spectral range from a purely Rayleigh-scattering atmosphere is also called the UV-AI (Torres et al., 1998), but the UV-AI is different from the AI as used in this review.



towards smaller particles, which makes it better suited as a proxy for CCN concentration at typical supersaturations than AOD.
For log-normal size distributions, AI is approximately proportional to the column aerosol number concentration (Nakajima
et al., 2001). Studies using models concluded that AI is a better predictor for CCN (Stier, 2016) and that AI – $N_d$ relationships
are better suited to predict $\Delta N_{d,ant}$ than AOD – $N_d$ relationships (Penner et al., 2011; Gryspeerdt et al., 2017).

Further refining this idea, Hasekamp et al. (2019a) aimed to retrieve the column CCN concentrations over oceans. The
analysis of polarimetric observations allowed to account for some aspects of the aerosol particle size distribution, and for
particle sphericity, which is related to particle hygroscopicity. This column-CCN retrieval implied larger $\bar{\beta}$, increasing the
resulting $RF_{aci}$ by almost 50%. It is an example of how additional information from polarimetry is useful for studying the CCN
to $N_d$ relationship.

However, neither the approach of Hasekamp et al. (2019a) nor the use of AI overcomes the problem of lack of horizontal
and vertical coincidence of CCN and $N_d$ retrievals. An option to overcome this problem is to make use of additional model
information. Satellite-retrieved AOD is assimilated into aerosol models e.g. in the Copernicus Atmosphere Monitoring Service
(Benedetti et al., 2009; Inness et al., 2019). The model predictions are applied to obtain aerosol information beneath clouds.
Such aerosol re-analysis information has been used for assessing $RF_{aci}$ in several studies (Bellouin et al., 2013; McCoy et al.,
2017; Bellouin et al., 2020a). However, assessing the validity of model results requires extensive and rigorous evaluation,
especially for coarsely-resolved models with regard to aerosol scavenging below clouds.

Yet another solution initially proposed by Feingold et al. (1998) and applied to satellite retrievals by Rosenfeld et al. (2016)
is to parameterize the cloud-base updraught, $w$, on the basis of cloud retrievals, rather than to retrieve the aerosol. For convec-
tive clouds, Zheng et al. (2015) suggested that $w$ scales with cloud-base altitude, which can be retrieved from satellites. For
stratocumulus clouds, Zheng et al. (2016) proposed that updraught is a function of cloud-top radiative cooling, and that this can
be computed by radiative transfer modelling on the basis of cloud quantities retrieved from passive sensors and thermodynamic
profiles from meteorological re-analyses. The retrieved profiles of $r_e$ together with deriving supersaturation as a function of $w$
and $N_d$ (Rosenfeld et al., 2016) then allows to parameterize the CCN concentration at any given supersaturation. This approach
does not suffer from the problem of lower detection limit. However, it has not yet been used to quantify the Twomey effect.

Concluding, all four approaches alleviate many problems encountered when using AOD. An ideal solution may be the
combination of several of these by assimilating, in addition to AOD, also polarimetric satellite observations, as well as lidar
measurements, into the analysis of the atmospheric state in high-resolution models.

## 3   Remote sensing of cloud droplet concentrations

The problem of the remotely sensed $N_d$ as used to estimate $\bar{\beta}$ has three different facets to it, which will be discussed in this
section, namely:

–   **Consideration of $r_e$ rather than $N_d$ in aerosol-cloud interaction studies:** In many studies, the droplet effective radius,
    $r_e$, is used, and the datasets are stratified with respect to $L$ in order to estimate $\bar{\beta}$. This is very difficult to perform
adequately and leads to biases.





– **Biases in the retrieved $N_{\mathrm{d}}$:** For the assessment of sensitivity, systematic (rather than random) errors in retrieved $N_{\mathrm{d}}$ are relevant. Also, $N_{\mathrm{d}}$ is not retrieved in standard operational procedures, so that inconsistencies between the retrieval of standard components and in the computation of $N_{\mathrm{d}}$ on the basis of retrievals can lead to additional errors.

– **Relationship of $N_{\mathrm{d}}$ formed at activation with retrieved and radiation-relevant $N_{\mathrm{d,top}}$:** Retrieved $N_{\mathrm{d,top}}$ refers to the drop concentration within the top 1 to 2 optical depths of the clouds, and it is $N_{\mathrm{d,top}}$ that is relevant for determining the cloud radiative effect. $N_{\mathrm{d}}$ sink processes such as coagulation imply that $N_{\mathrm{d,top}}$ is smaller than the one resulting from activation at of above cloud base, $N_{\mathrm{d,base}}$.

$N_{\mathrm{d}}$ is vertically constant for single-layer, purely-liquid-water clouds with (i) a vertically homogeneous droplet size spectrum, (ii) for adiabatically stratified clouds, or (iii) for sub-adiabatic clouds in which mixing is homogeneous. However, in many situations, precipitation formation or entrainment can lead to reduction of $N_{\mathrm{d}}$ above cloud base. In such situations, it is $N_{\mathrm{d,top}}$ that is relevant to determine the cloud radiative effect (cloud albedo in Eq. 2). Building on Eq. 4 thus gives

$$\overline{\Delta N_{\mathrm{d,top,ant}}} = \frac{\mathrm{d}\overline{N_{\mathrm{d,top}}}}{\mathrm{d}\overline{N_{\mathrm{d,base}}}} \cdot \overline{\left[\int\limits_{w=-\infty}^{\infty} \left.\frac{\partial N_{\mathrm{d,base}}}{\partial \ln a}\right|_{w} \mathcal{P}(w)\,\mathcal{P}(a)\,\mathrm{d}w\right]} \cdot \overline{\Delta \ln a_{\mathrm{ant}}} = \hat{\beta} \cdot \overline{\Delta \ln a_{\mathrm{ant}}}. \tag{5}$$

When estimating $\bar{\beta}$ as regression coefficient from, e.g. satellite-retrieved $N_{\mathrm{d}}$ and a proxy for CCN such as AOD, it is thus this $\hat{\beta}$ that is inferred.

## 3.1   Considering $r_{\mathrm{e}}$ rather than $N_{\mathrm{d}}$

Many past studies have used operationally-retrieved $r_{\mathrm{e}}$ rather than $N_{\mathrm{d}}$ in aerosol-cloud interaction studies. However, $r_{\mathrm{e}}$ is a function of both $N_{\mathrm{d}}$ and $L$. This introduces the requirement for stratifying the data with respect to $L$ in order to estimate $\hat{\beta}$. To further complicate matters, $N_{\mathrm{d}}$ and $L$ have been found to be correlated (e.g. Michibata et al., 2016; Gryspeerdt et al., 2019). A precise estimation of $\hat{\beta}$ is thus only possible for a large amount of data combined with suitable binning by $L$. Errors in this approach that are related to a lack of data increase at aggregated scales (McComiskey and Feingold, 2012). Using derived $N_{\mathrm{d}}$ is therefore preferable to avoid unnecessary complications.

## 3.2   Biases in the $N_{\mathrm{d}}$ retrieval

Satellite retrievals of $N_{\mathrm{d}}$ were extensively reviewed by Grosvenor et al. (2018). Since $N_{\mathrm{d}}$ currently is not retrieved by operational algorithms and new developments to retrieve $N_{\mathrm{d}}$ (e.g. from polarimetry) are still in their infancy, the most frequently used method is to infer $N_{\mathrm{d}}$ from retrieved $r_{\mathrm{e}}$ and cloud optical depth, $\tau_{\mathrm{c}}$, using the relationship

$$N_{\mathrm{d}} = \gamma \cdot \tau_{\mathrm{c}}^{\frac{1}{2}} \cdot r_{\mathrm{e}}^{-\frac{5}{2}} \tag{6}$$

where $\gamma \approx 1.37 \cdot 10^{-5}\,\mathrm{m}^{-0.5}$ is a parameter provided as a constant here but more realistically depending on many uncertain properties such as the vertical profile of effective radius and liquid water content and the droplet size distribution (Boers et al., 2006; Quaas et al., 2006; Grosvenor et al., 2018). The relationship in Eq. 6 assumes that clouds are adiabatic or nearly





adiabatic (i.e. adiabatic clouds or sub-adiabatic clouds with homogeneous mixing only; Brenguier et al., 2000). The most common method uses a bispectral approach to retrieve $r_e$ and $\tau_c$ (Nakajima and King, 1990). Various error sources lead to an overall retrieval error for $N_d$ (Grosvenor et al., 2018; Wolf et al., 2019). As can be deduced form Eq. 6, the most important contributions are from retrieval errors in $r_e$. Other error sources are the uncertainty in sub-adiabatic factor, the cloud model used in the retrieval, and the droplet size distribution width. Satellite retrievals of the vertical profile of cloud droplet size may

help to improve the retrieval (Chang and Li, 2002; Chen et al., 2008). Grosvenor et al. (2018) identified biases of retrieved $N_d$ especially for broken cloud regimes and at large solar zenith angles. In stratocumulus, it was suggested that the retrieval yields the most trustworthy results when considering only the brightest pixels (Zhu et al., 2018). For the ideal case of homogeneous, low-latitude stratiform clouds, relative errors in the $N_d$ retrieval at pixel scale are quantified as 78% (Grosvenor et al., 2018). In such cases, the error was assumed as random. However, systematic errors occur especially in broken cloud regimes and

for large solar zenith angles, leading to an underestimation (broken cloudiness) and overestimation (large solar zenith angles), respectively, of $N_d$.

    For improvements in estimates of $N_d$, it would be beneficial to formulate a retrieval in terms of $N_d$ directly rather than in terms of $r_e$ and $\tau_c$. It is also possible to reduce uncertainties in retrievals of $r_e$ and $\tau_c$, or to reduce uncertainties related to assumptions of the vertical structure of the cloud and particle size distribution shape. Approaches to quantify and partly correct

for retrieval biases as discussed in Grosvenor et al. (2018) include accounting for cloud heterogeneity by using those channels in passive imagers that provide spatial resolution that exceeds the one at which the standard retrieval products are provided. The combination of passive observations with radar may further improve the retrieval (Posselt et al., 2017). Substantially more accurate retrievals of $r_e$ and additional relevant information about droplet size distributions may also come from multi-angular polarimetric measurements (Alexandrov et al., 2012b, a; Shang et al., 2019), which will be possible from orbit at pixel

level from the Hyper-Angular Rainbow Polarimeter-2 (HARP-2) on the NASA PACE mission (Martins et al., 2018; McBride et al., 2019). Polarimetric retrievals allow to infer the spectral width or general shape of the droplet size distribution at cloud top (Hu et al., 2007). This approach is not substantially sensitive to sub-pixel cloudiness, mixed-phase conditions and 3D radiative effects (Alexandrov et al., 2012b). The sensitivity of derived $N_d$ to uncertainties in $r_e$ from polarimetric retrievals may further be reduced by additionally inferring cloud physical thickness. In this case, $N_d$ can be inferred as linear in $\tau_c$

and inversely linear in geometrical thickness and mean droplet extinction cross-section at cloud top (Sinclair et al., 2019). The geometrical thickness can also be inferred from total and/or polarized reflectances measured in oxygen absorption bands (Sanghavi et al., 2015; Richardson et al., 2019) or by retrieving cloud base using lidar (Mülmenstädt et al., 2018) or using multi-angle observations (Böhm et al., 2019). When exploiting passive observations together with lidar, $N_d$ at cloud top can be robustly inferred as the ratio of in-cloud extinction (lidar) and extinction cross section (passive). A slightly less direct approach

using depolarization to estimate extinction and effective radius to estimate extinction cross section has been presented by Hu et al. (2007).





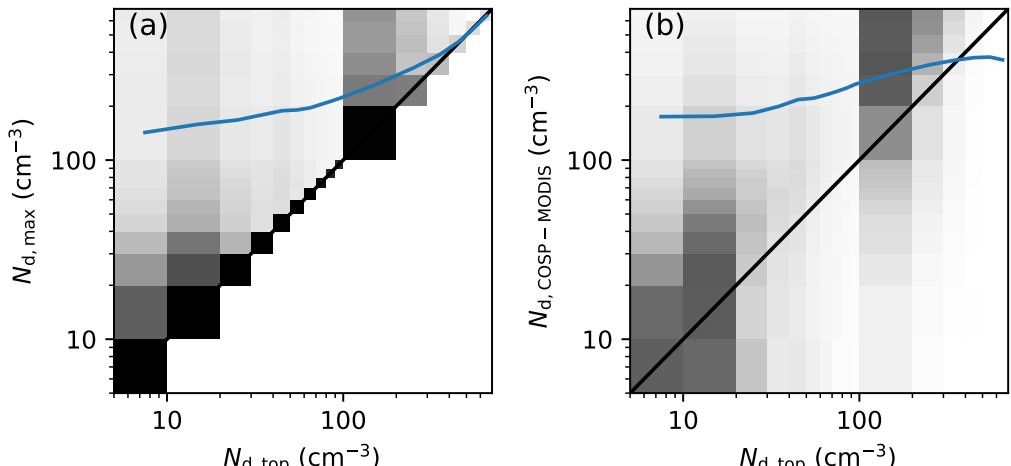

**Figure 1.** Analysis of $N_d$ in the "virtual reality" of a cloud-resolving simulation: Droplet number concentration ($cm^{-3}$) from the ICON large-eddy simulation (156 m horizontal resolution) over the domain of Germany for 2 May 2013 (Heinze et al., 2017), for the overpass times of the Terra and Aqua satellites for which the swath of the MODIS instrument covered the domain (twice around 10:30 h local solar time for Terra, twice around 13:30 h for Aqua) even if no actual data are used in this analysis (Costa-Surós et al., 2019). Joint histograms, normalized along the y-axis as in Gryspeerdt et al. (2016) for (a) column-maximum (proxy for activated CCN) vs. cloud-top $N_d$ (taken at $\tau_c = 1$ integrated from cloud top) and (b) $N_d$ derived from $r_e$ and $\tau_c$ as in Grosvenor et al. (2018) vs. cloud-top $N_d$, where both quantities are computed as seen from a satellite using COSP (Bodas-Salcedo et al., 2011). The blue line is the mean in each bin for cloud-top $N_d$.

### 3.3 Relationship between $N_d$ formed at CCN activation and retrieved radiation-relevant $N_d$

In stratiform clouds, droplets form at cloud base which is where $N_d$ most closely relates to CCN. In convective clouds, updraught in some cases increases with height above cloud base. Hence, additional CCN may activate above cloud base and lead to vertically increasing $N_d$ in the lower third of the cloud with a decrease further up (Endo et al., 2015). However, in most cumulus, and in stratiform clouds, $N_d$ is found to be largest at cloud base and to slightly decrease above it (Jiang et al., 2008; Small et al., 2009; vanZanten et al., 2011). In the approach discussed by Grosvenor et al. (2018), the retrieved $N_d$ is representative of the cloud-top reflectance, and thus, the relevant proxy for the $N_d$ that matters for cloud albedo and $RF_{aci}$ (Platnick, 2000). To which extent the microphysical structure of lower parts of a cloud exactly impacts radiation (weighting function) depends on the multiple scattering and thus on the vertical structure of $N_d$ itself (Platnick, 2000; Krisna et al., 2018). For vertically constant $N_d$, the retrieved $N_d$ represents the droplet concentration formed by CCN activation. However, there are $N_d$ sinks, in particular due to collision-coalescence (in liquid clouds, the autoconversion and accretion, or "warm rain" processes) that lead to droplet depletion. Wood (2006) demonstrated that the depletion is exponential in precipitation rate and estimated a loss in $N_d$ of $100\,cm^{-3}\,day^{-1}$ for precipitation rates of $1\,mm\,day^{-1}$. There may also be lateral and vertical mixing (of heterogeneous type, Lehmann et al., 2009) of cloud air with environmental cloud-free air that can lead to the full evaporation





of droplets. In both sinks for $N_\mathrm{d}$, the one due to precipitation formation and the one due to mixing, the retrieved $N_\mathrm{d}$ is expected to be smaller than the $N_\mathrm{d}$ formed at activation of CCN. In an aged cloud, however, updraughts may have decayed such that no additional droplets are formed, while existing droplets persist, or may be advected from elsewhere. Also large raindrops may break up into droplets, in which case $N_\mathrm{d}$ is increased. Arguably, it is the right choice to relate the retrieved $N_\mathrm{d}$, as the

radiation-relevant one, to CCN, i.e. to use $\hat{\beta}$, when computing the $N_\mathrm{d}$ to CCN sensitivity with the aim to constrain $\mathrm{RF_{aci}}$.

Cloud-resolving models are a good tool to investigate these interpretations (McComiskey and Feingold, 2012). Fig. 1 shows an analysis of a large-domain large-eddy simulation with the ICON-LEM model (Heinze et al., 2017; Costa-Surós et al., 2019). CCN concentrations in these simulations are relaxed towards pre-computed spatially and temporally varying fields and are consumed at activation. In the 22 million grid columns, the droplet concentration at cloud top (what is retrieved from satellites)

is compared to the maximum droplet concentration (approximately the concentration of activated CCN / formed droplets). This demonstrates that there is a link between the droplet concentration formed at activation and $N_\mathrm{d}$ determining the cloud radiative effect at its top. These two quantities correlate rather well in the joint histogram, though that link is far from one-to-one. The second plot (Fig. 1b) assesses the possibility to infer cloud-top $N_\mathrm{d}$ from cloud-top $r_\mathrm{e}$ and $\tau_\mathrm{c}$ (Grosvenor et al., 2018). For this, the MODIS simulator (Pincus et al., 2012) that is part of the Cloud Feedback Model Intercomparison Project (CFMIP)

Observational Simulator Package (COSP; Bodas-Salcedo et al., 2011) is applied to the model output to compute cloud-top $r_\mathrm{e}$ and $\tau_\mathrm{c}$. From these, $N_\mathrm{d}$ is computed as in Eq. 6. This approach mimics the satellite retrieval but assumes no retrieval errors, i.e. the comparison is a lower bound on the accuracy of the retrieved $N_\mathrm{d}$ in representing the actual $N_\mathrm{d}$ at cloud top. There is a meaningful co-variation of the two quantities, but it is far from perfect. In particular, there is a systematic overestimation of $N_\mathrm{d}$ in the retrieval approach, especially at low $N_\mathrm{d}$. The relative error even is a function of $N_\mathrm{d}$, with larger relative errors at low

$N_\mathrm{d}$.

In conclusion, the fact that cloud-top $N_\mathrm{d}$ is in general lower than $N_\mathrm{d}$ at activation height implies that $\hat{\beta}$ is smaller than unity. This is not a problem, but a desired analysis result when studying the Twomey effect. However, $N_\mathrm{d}$ obtained from retrieval products is biased high for low values of $N_\mathrm{d,top}$. This relative error, which is a function of $N_\mathrm{d}$, implies that the regression between satellite-derived $N_\mathrm{d}$ and CCN yields a sensitivity that is too weak.

**4   Cloud regime dependence**

Aerosol-cloud interactions depend on cloud regime (Stevens and Feingold, 2009; Mülmenstädt and Feingold, 2018). When it comes to $\mathrm{RF_{aci}}$, there are three reasons for this: (i) the radiative sensitivity (Oreopoulos and Platnick, 2008; Alterskjær et al., 2012), i.e. the first two terms on the right-hand-side of Eq. 2 (in particular the sensitivity expressed in Eq. 1), (ii) the updraught-dependence of $\hat{\beta}$, and (iii) the dependence of the relation of cloud-top to cloud-base $N_\mathrm{d}$ on characteristics of turbulence and

rain. The latter two are of interest here. "Cloud regime" thus means here, a cluster of clouds with similar $\mathcal{P}(w)$ and similar $\frac{\mathrm{d}\overline{N_\mathrm{d,top}}}{\mathrm{d}\overline{N_\mathrm{d,base}}}$ in Eq. 5. When considering CCN at a certain supersaturation level, $\hat{\beta}$ is larger at larger updraught, $w$ (MacDonald et al., 2020). Broadly, cumulus clouds have larger $w$ than stratiform clouds. In addition, clouds over land usually have larger $w$





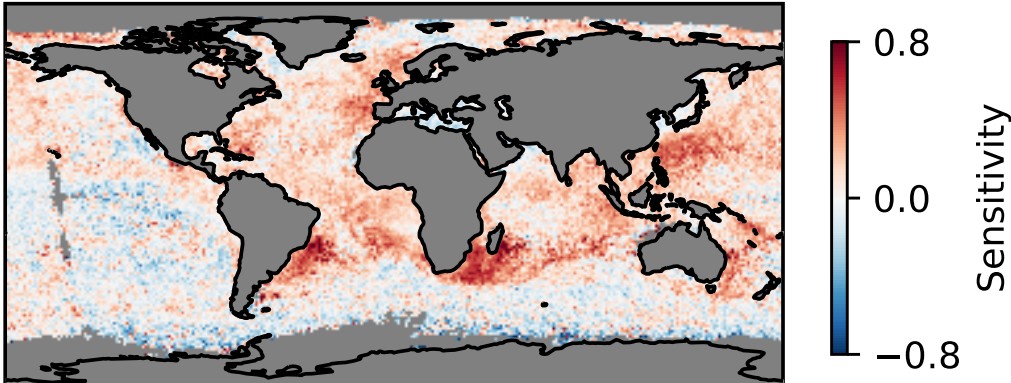

**Figure 2.** Regression coefficients of $N_d$ computed on the basis of retrievals of the MODerate Resolution Imaging Spectroradiometer (MODIS; Platnick et al., 2017) as in Grosvenor et al. (2018) and AI from MODIS (Levy et al., 2013) from the daily temporal variability in grid-boxes of $1° \times 1°$.

than clouds over ocean. Building on Eq. 5, this suggests a regime-based analysis expressed as

$$\overline{\Delta N_{d,top,ant}} = \left. \frac{d\overline{N_{d,top}}}{d\overline{a}} \right|_{regime} \cdot \overline{\Delta \ln a_{ant}}. \tag{7}$$

Fig. 2 shows the spatial distribution of the $N_d$ − AI regression coefficient from its temporal variability within $1° \times 1°$ grid boxes. The large spatial heterogeneity is not straightforward to interpret. Some problems may be due to the lack of aerosol retrieval sensitivity (e.g. in regions with low CCN concentrations such as the southern oceans) or lack of vertical or horizontal co-incidence (e.g. in regions with heterogeneous aerosol and large cloud coverage such as mid-latitude storm tracks). However, aspects of the geographical heterogeneity may indeed be attributable to physical and relevant reasons. However, it is difficult to

determine any attributable factors in the spatial and cloud regime variations in $\hat{\beta}$ (Gryspeerdt and Stier, 2012) before retrieval errors are remedied.

In precipitating situations, the two-way interactions can lead to large challenges in determining the $\hat{\beta}$ term (Ekman et al., 2011). Precipitation scavenges aerosol and, in certain situations, the interplay between aerosol, droplet concentrations and precipitation determines both aerosol and droplet concentrations. This may yield bifurcations between situations with large $N_d$

in which no drizzle forms, and very low $N_d$ and cloud dissolution when precipitation forms (e.g. Yamaguchi et al., 2017). In such situations, it is particularly challenging to identify the $N_d$ − CCN concentration sensitivity.

## 5   Aggregation scale

The impact of aggregation scale on estimates of $\beta$ has been discussed in detail by McComiskey and Feingold (2012). Their key conclusion is that at scales larger than the cloud variability scale of about 1 to 10 km, aerosol and cloud data become





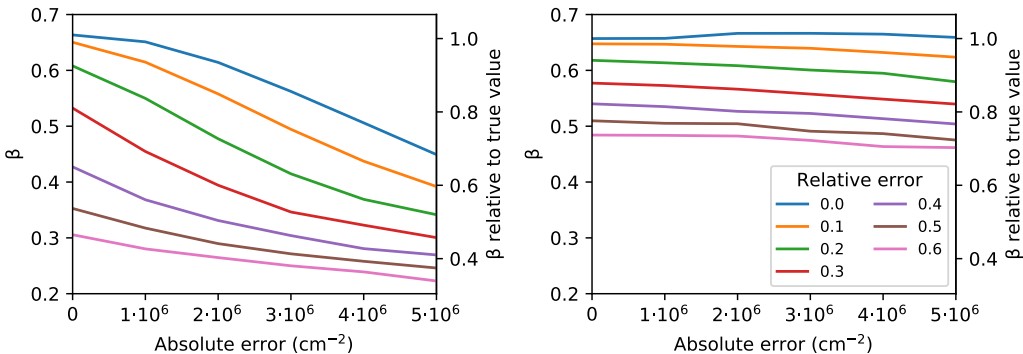

**Figure 3.** $N_{\mathrm{d}}$ – column CCN sensitivity as a function of the stochastic error in column CCN (absolute additive error) in an emulated analysis as in Hasekamp et al. (2019a), for different relative (multiplicative) errors, for (left) the full range of data, including low $N_{\mathrm{CCN}}$ values and (right) excluding $N_{\mathrm{CCN}} < 10^{7}\,\mathrm{cm}^{-2}$. Hasekamp et al. (2019a) suggest a realistic error is about $0.2 \cdot N_{\mathrm{CCN}} + 4 \cdot 10^{6}\,\mathrm{cm}^{-2}$.

de-correlated so that the diagnosed $\beta$ becomes less and less representative for individual cloud parcels. In turn, Sekiguchi et al. (2003) computed $\hat{\beta}$ for different aggregation scales and demonstrated that it actually increases with larger scales. An analysis of spatio-temporal vs. temporal-only co-variability of $N_{\mathrm{d}}$ and AOD by Grandey and Stier (2010) found that $\hat{\beta}$ is larger when considering spatio-temporal variability over entire regions compared to only temporal variability at individual $1° \times 1°$ grid boxes. These results are opposite to those expected from the process-based conclusions of McComiskey and Feingold (2012).

A possible problem in the Sekiguchi et al. (2003) study is their use of $r_{\mathrm{e}}$ rather than $N_{\mathrm{d}}$ and the subsequent need to stratify by $L$. McComiskey and Feingold (2012) demonstrated that this approach becomes more problematic with increasing aggregation scale. However, their analysis suggested a low-bias in $\beta$ at coarser scales due to stratification by $L$. Reduced $\hat{\beta}$ at small scales could occur if aerosol conditions become too small to diagnose the full range of co-variability due to smaller sample sizes at smaller scales.

Concluding, from a process point of view, aggregation over larger scales is expected to lead to a decrease in estimated $\hat{\beta}$. In turn, to study the large-scale Twomey effect, an aggregate $N_{\mathrm{d}}$ – CCN relationship is desired as it is the large-scale $\Delta N_{\mathrm{d,ant}}$ that matters for the radiation perturbation and because the anthropogenic aerosol perturbation can only be inferred at a large scale. The often adopted choice of a $1° \times 1°$ gridding is somewhat motivated by the suggestion that this is a scale at which aerosol concentrations are considered homogeneous (Anderson et al., 2003) and loosely (to within a factor of about 2 in each

horizontal direction; re-analyses are more at a $50\,\mathrm{km}$ scale, many general circulation models still are as coarse as $200\,\mathrm{km}$) related to the scale at which models infer the anthropogenic perturbation of CCN. A rigorous study on the scale-dependency of $\hat{\beta}$ and the consequences thereof for $\mathrm{RF}_{\mathrm{aci}}$ would be desirable.



## 6 Quantification for the regression coefficient

When sensitivities are approximated by linear regression coefficients from an ordinary least squares (OLS) line fitting method,
rather than derived e.g., in form of joint histograms, the problem of regression dilution arises to the extent that the aerosol
quantity shows errors: the regression coefficient becomes gradually smaller as the stochastic error increases (Pitkänen et al.,
2016). This is quantified for the column-CCN vs. $N_d$ sensitivity evaluated as regression coefficient in Fig. 3. Due to the
regression dilution, the sensitivity decreases by factors of 2 to 3 as the error in column CCN increases when considering
relative errors of 50%. This can to a large extend be remedied by ignoring data points at low CCN concentrations from the
regression (Fig. 3, right panel). However, this solution is limited to regions not dominated by low aerosol concentrations. Fig. 3
also illustrates that an absolute bias in the data translates to relative bias in logarithmic scale. Therefore, if no bias correction
is applied, an absolute bias in the data will cause a bias in the sensitivity estimates. As shown by Pitkänen et al. (2016), the
regression dilution in turn becomes weaker at coarser aggregation scales in cases of auto-correlated data, which is the case for
aerosol concentrations. This is of relevance in case of both temporal and spatial aggregation. In other words, the systematic low
bias in the sensitivity is reduced if data are aggregated. This could partly explain some previous findings of increasing sensitivity
with decreasing resolution (see discussion in the previous section), in addition to the actual bias due to the aggregation over
smaller scale of cloud processes. These considerations imply that it is necessary to either analyse the full variability of aerosol-
cloud interactions, e.g. in the form of joint histograms, or to account for the regression dilution using established mathematical
approaches that properly consider measurement uncertainties, as discussed in Mikkonen et al. (2019), for instance.

## 7 Conclusions

The radiative forcing due to aerosol-cloud interactions, or Twomey effect, requires quantification based on observational data,
since models are associated with large uncertainties. At a large scale, this calls for satellite retrievals. There are, however, large
challenges when using satellite data and this review summarizes these challenges and suggests some potential ways forward.
The key data-related question is the sensitivity of droplet concentration, $N_d$, to perturbations in the cloud-active aerosol, i.e. the
cloud condensation nuclei (CCN) concentration at or above cloud base. The most widely-used proxy of the cloud-base CCN
concentration is the aerosol optical depth (AOD), or alternatively the aerosol index (AI), taken from cloud-free pixels in the
vicinity of the locations the cloud retrievals. The four main caveats with AOD are the lack of vertical resolution, the additional
influence of hygroscopic swelling, the fact that the detected aerosol might be not active as CCN, as well as the impossibility to
retrieve it in cloudy skies. In terms of the vertical resolution, satellite-based lidar offers help. However, current lidar retrievals
are even more constrained to large aerosol concentrations than passive AOD retrievals. EarthCARE's ATLID lidar will allow
direct inference of the ratio of backscatter to extinction, enabling greatly improved retrievals of aerosol extinction profile.
Adding a second wavelength with ATLID capabilities and combining it with polarimetric measurements would substantially
extend vertically resolved aerosol information content. In terms of horizontal co-location, trajectory computations may help to
identify the aerosol representative of that affecting specific clouds. However, this requires extra effort and reliable information





about trajectories. The hygroscopic swelling can be addressed by parameterisations on top of the retrievals. Further relevant information is possible from polarimetric measurements.

Cloud droplet number concentration, $N_\mathrm{d}$, is only indirectly available from current operational satellite retrievals. It is generally computed from retrieved cloud-top droplet effective radius, $r_\mathrm{e}$ and cloud optical thickness, $\tau_\mathrm{c}$, leading to substantial biases in comparison to the cloud-top droplet number concentration, especially in inhomogeneous, broken and/or precipitating

cloud regimes. Sink processes for $N_\mathrm{d}$ and variability due to atmospheric dynamics, including turbulent mixing, imply that the radiatively-relevant cloud-top $N_\mathrm{d}$ far from perfectly relates to the $N_\mathrm{d}$ formed by CCN activation. In addition, at a given CCN concentration, the updraught variability also leads to sensitivies of $N_\mathrm{d}$ to CCN that are much less than one-to-one. These latter two facts are not problematic when assessing the $N_\mathrm{d}$ to aerosol sensitivity from data for the estimation of the Twomey effect. In fact, it is desirable to quantify at a large scale the net impact of aerosol perturbations of the (radiatively-relevant) cloud-top

$N_\mathrm{d}$ that accounts for updraught and $N_\mathrm{d}$ sink variability. However, it is necessary to operationally retrieve $N_\mathrm{d}$, rather than to indirectly compute it from $r_\mathrm{e}$ and $\tau_\mathrm{c}$ retrievals. It is also necessary to improve these retrievals in particular for low droplet concentrations and broken cloud conditions. In addition, these retrievals should take into account additional information e.g. about the onset of drizzle.

Regression dilution influences the statistically inferred sensitivity as a result of stochastic retrieval errors in CCN concen-

tration. On the one hand, at aggregate scales, this problem becomes less relevant due to the autocorrelation of the aerosol concentrations. The relationship between $N_\mathrm{d}$, that varies at cloud-dynamics scales, and CCN proxies becomes weaker at aggregate scales. Relative retrieval errors in $N_\mathrm{d}$ that depend on actual $N_\mathrm{d}$ (with larger high-biases at low true $N_\mathrm{d}$) lead to a further reduction in the estimated sensitivity. It is thus necessary to account for the impact of CCN errors in the statistics and to optimize the resolution of $N_\mathrm{d}$ and CCN retrievals towards cloud-scale resolutions.

The recent study by Hasekamp et al. (2019a) made use of polarimetric satellite measurements to suggest a global-ocean average $N_\mathrm{d}$ to CCN sensitivity of 0.66. This, combined with anthropogenic column-CCN concentrations and radiative sensitivities, translates into a global Twomey effect of -1.1 W m$^{-2}$. The net effect of the remaining problems laid out above suggests that this likely is still too low an estimate for the $N_\mathrm{d}$ – CCN sensitivity, implying a stronger Twomey effect. However, it is desirable to add the extra steps to improve the data-tied quantification for process understanding as well as for evaluating and

improving climate models.

In situ and ground-based observations, as well as analysis of cloud-resolving dynamical models, may be a path forward for the evaluation of critical aspects in the satellite-based analysis. Important steps would be the quantification of updraught PDFs for different cloud regimes, and the assessment of horizontal homogeneity of aerosol concentrations.

*Author contributions.* J.Q. led the writing of the manuscript with significant contributions from all authors.

*Competing interests.* The authors declare there are no competing interests.





*Acknowledgements.* The work of J.Q., A.E., A.N. and P.S. was supported by the European Union via its Horizon 2020 project FORCeS (GA 821205). The work of A.N. was further supported by the European Research Council via the project PyroTRACH (ERC-2016-COG) funded from H2020-EU.1.1. - Excellent Science (project ID 726165). This review originated from discussions at the 2019 Nanjing workshop of the Aerosols-clouds-precipitation and climate (ACPC) initiative (acpcinitiative.org) and benefited from discussions within the group

"Study of aerosol–cloud interactions based on satellite observations of the terrestrial underlying surface–atmosphere system: a new frontier of atmospheric science", hosted by the International Space Science Institute (ISSI). We thank the German Climate Computing Centre (Deutsches Klimarechenzentrum, DRKZ) and the research programme "HD(CP)$^2$ - High Definition Clouds and Precipitation for Climate Prediction" funded by the German Federal Ministry of Education and Research (BMBF) within the framework programme "Research for Sustainable Development (FONA)", www.fona.de, for making the ICON-LEM simulations available. J.Q. is grateful to the NASA Goddard Institute

for Space Studies, New York, for hospitality during a research stay. We thank Andrew Ackerman and Patrick Chuang for constructive discussions.



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
