# Peer review of "Constraining the Twomey effect from satellite observations: Issues and perspectives"

_Atmospheric Chemistry and Physics, 2020_

## Referee Comment (RC1) · Anonymous Referee #3 · 12 Jun 2020

This overview paper is a pretty substantial and concise overview of Twomey effect diagnostics from space, principally with passive solar observations. The paper is generally well-written (save a few passages – something not unexpected given the many co-authors and the unavoidable mixing of styles) and breaks down the problem in an intelligent and intuitive manner. The heart of the paper is eq. (4) which is then further recast as eq. (5). These equations indicate that assessing the strength of the Twomey effect rests on being able to predict the change in cloud droplet number concentration given an anthropogenic CCN perturbation. The latter is not examined; rather the paper focuses on whether the sensitivity of droplet concentration to changes in CCN can be inferred from space observations. The issues investigated are whether aerosols (and what aerosols in terms of vertical location) can stand-in for CCN and at which

level in the cloud the knowledge of the droplet concentration is relevant to calculate the Twomey radiative perturbation. Given the nature of the paper, there is really no original research, but there is plenty of good insight. The paper lacks visual support: there are only three figures in 18 pages. To me at least, it seemed as if the paper loses steam starting in section 4 when text appears to suffer from deteriorating clarity and appears to be more hastily written. But all in all, this is a very noteworthy effort that does not need much of a revision before it becomes a reference to be frequently visited by the aerosol-cloud interaction community.

Some remarks/suggested edits:

Line 10 and many instances thereafter: "vertical wind" does not seem the right term; rather people traditionally use the term "updraft velocity", or, given the convention of this paper, "updraught velocity".

Line 11: "10s", this read like 10 seconds to me, so better write explicitly "tens".

Line 21: "the impossibility" (of retrieving base CCN): Well, some would disagree, and the paper itself does cite Rosenfeld et al. (2016) who claim that such retrieval is possible. See line 289.

Line 53: Cloud horizontal extent is actually irrelevant, if the quantity of interest is cloud albedo. Cloud fraction becomes relevant only when the dependences of the Twomey effect on spatial scales is discussed and then only when mixtures of clear and cloudy skies are considered, namely the Twomey effect is expressed in terms of the cloud radiative effect.

Line 54: "$a_c$ is a monotonic function of $N_d$": only when the cloud condensate is constant.

Eq. (2): A derivative of absolute $a_c$ change with respect to a relative (logarithmic) $N_d$ change is shown, while eq. (1) is expressed in terms of relative changes for both quantities. It may make sense to keep these consistent. See also line 81.

Line 66: SOLAR zenith angles.

Lines 75-79: N_d is also a function of L (you say that actually in line 323), so I don't understand the argument here, which is fundamental for insisting that Twomey effect studies are conducted in terms of N_d (not a directly retrievable quantity) and not r_e (which is directly retrieved). Changes in L can be distributed as both droplet size and droplet number changes, no? See also lines 435-436 about the need to stratify by L when using r_e.

Lines 169-171: Need to clarify that this is the case for passive SWIR observations. Lidar retrievals are discussed elsewhere in the paper.

Line 200: I suggest "become less representative of aerosol variability".

Line 201: To be consistent with elsewhere in the text: "updraughts".

Lines 271-272: It is implied here that AI is routinely available from space. Is it? For example, MODIS dark target provides AI only over ocean. Is it reliably retrieved? Fig. 2 excludes the land, probably because of this exact unavailability of AI over continents.

Line 284: The MERRA-2 aerosol re-analysis is also another popular product. Lates in lines 287-288, it is not clear how one can evaluate re-analysis aerosol, especially underneath cloud. One has to use observations that are not part of the assimilation process.

Line 294: I suggest "derivations of supersaturation".

P. 12 discussion on N_d retrieval uncertainties: The discussion seem to suggest that higher resolution measurements are needed to reduce cloud heterogeneity effects, yet the retrievals should eventually be coarsened anyway to reduce the random error.

Lines 359 and 362: Deriving cloud base and cloud physical thickness is of course one of the most difficult problems in space-based remote sensing. Lidar can be useful only when the clouds are optically thin (optical thickness below 3-4). So, I wouldn't count

too much on space-based lidars for many of the clouds that are relevant to the Twomey effect.

Line 401: "beta_hat is smaller than unity". Earlier, line 87, it was established that beta is smaller than unity. No range was given for beta_bar, but presumably the same implies. Do the authors then mean to say in line 401 that beta_hat is smaller than beta_bar?

Line 438: conditions cannot become small, so the authors need to rephrase.

Line 445: I suggest you say "closer to ∼50 km scales".

Section 6: I found this section about confusing, but I think mostly because of my unfamiliarity with the "regression dilution" concept and the ways its impact is assessed. The term does indeed exist and describes the biasing of the regression slope towards zero values, but you may want to provide a brief definition and description. For people who are familiar with this bias tendency this section may make more sense. Please revisit and ensure that you provide maximum clarity to the uninitiated.

Lines 473-474: "the impossibility to retrieve it in cloudy skies". This is a sweeping statement which need some qualifiers. Yes, you can't probably retrieve aerosol under clouds in most situations, but with lidar it is possible both above and below clouds for certain clouds. Also you can retrieve aerosol between individual clouds of a cloud field from both passive and active. Such a cloud field is still "cloudy skies".

Line 480: I suggest "in addition to retrievals".

Line 486: I suggest "relates imperfectly to the N_d".

Line 487: You mean sensitivities less than one? I don't understand as it is currently written.

Line 504: I suggest "quantification supported by data".

---

## Short Comment (SC1) · 8 Jul 2020

Dear authors, I would like to draw your attention to a recently published ACP paper that makes use of vertically resolved CALIPSO retrievals for investigating co-variability between cloud droplet number concentration (Nd) from MODIS and aerosols (Painemal et al., 2020). We also discuss the advantages of using vertically resolved aerosol properties relative to the common approach of using aerosol optical depth. The material discussed in Painemal et al. (2020) could be relevant to the topic discussed in your manuscript. https://www.atmos-chem-phys.net/20/7167/2020/acp-20-7167-2020.html

Best regards David Painemal SSAI/ NASA Langley Research Center david.painemal@nasa.gov

---

## Referee Comment (RC2) · Anonymous Referee #1 · 13 Aug 2020

General comment:

This article provides a comprehensive review on how to estimate the Twomey effect from satellite observations. The review builds upon simple formulations that decompose the radiative forcing due to the Twomey effect into several terms corresponding to different physical processes accounting for spatial (horizontal) and temporal variabilities of cloud, aerosol and dynamical fields, as represented by Equations (2), (3) and (4). These equations well serve as a basis for discussing and pointing out issues in quantifying the Twomey effect at a scale relevant to climate, which is of particular interest in this review. Key sources of error or uncertainty in quantifying the Twomey effect are then reasonably identified and separated to facilitate the discussion and propose way forward for alleviating the overall uncertainty. I only have relatively minor com-

ments that I would propose for the authors to consider for further improvement of the manuscript.

Specific comments:

1. This may be just my misunderstanding, but the authors seem to argue that a use of Nd, instead of Reff, can circumvent constraining LWP for quantifying the Twomey effect. Is it correct? To my understanding, estimates of the Twomey effect, by its definition, always require the LWP to be constant so that the data always need to be stratified by LWP whether Nd or Reff is used for analysis. Can the authors clarify why Nd is more advantageous than Reff for estimating the Twomey effect? Explanations in Section 3.1 are not convincing enough.

2. The authors show several lines of evidence that past studies likely underestimated the radiative forcing due to the Twomey effect with some quantitative information of how large is the underestimates (such as those shown in Figures 1 and 3). I am just wondering if the authors could propose a range of estimate for the radiative forcing that is "corrected" from the existing estimate (like IPCC AR5) accounting for the factors listed in the manuscript that may have caused the underestimate. Such a quantitative estimate would be desirable to show if it is possible.

3. In section 2.1, the authors should explain in more detail why and how the EarthCARE lidar can improve the accuracy of retrieving and discriminating aerosols and clouds, particularly for those of readers who are not familiar with EarthCARE lidar specification. In particular, more explanations would be useful for how ATLID can (i) better distinguish the optically thin clouds and aerosols and (ii) better profile the aerosol extinction, with the capability of HSRL enhanced from CALIOP.

4. In section 2.2: How can recent geostationary satellites with unprecedentedly high spatial and temporal resolutions provide potentially useful information for horizontal collocation in the context of trajectory approach? For instance, Kikuchi et al. (2018) exploited the high frequency sampling of Himawari-8 to create a new data set of AOD

interpolated to the location collocated with clouds that is likely more relevant to CCN.

5. In section 2.3: Is there any specific way of parameterizing the dry aerosol properties from the humidified one? Some literature information would be desirable to let the readers to have more specific ideas of the issue of swelling.

Reference:

Kikuchi, M., and Coauthors, 2018: Improved hourly estimates of aerosol optical thickness using spatiotemporal variability derived from Himawari-8 geostationary satellite. IEEE Trans. Geosci. Rem. Sens., 56, 3442-3455, doi:10.1109/TGRS.2018.2800060.

---

## Author Comment (AC1) · 24 Sep 2020

General comment:
This article provides a comprehensive review on how to estimate the Twomey effect from satellite observations. The review builds upon simple formulations that decompose the radiative forcing due to the Twomey effect into several terms corresponding to different physical processes accounting for spatial (horizontal) and temporal variabilities of cloud, aerosol and dynamical fields, as represented by Equations (2), (3) and (4). These equations well serve as a basis for discussing and pointing out issues in quantifying the Twomey effect at a scale relevant to climate, which is of particular interest in this review. Key sources of error or uncertainty in quantifying the Twomey effect are then reasonably identified and separated to facilitate the discussion and propose

way forward for alleviating the overall uncertainty. I only have relatively minor comments that I would propose for the authors to consider for further improvement of the manuscript.

*We thank the reviewer for their excellent summary and kind assessment of the manuscript.*

Specific comments:

1. This may be just my misunderstanding, but the authors seem to argue that a use of Nd, instead of Reff, can circumvent constraining LWP for quantifying the Twomey effect. Is it correct? To my understanding, estimates of the Twomey effect, by its definition, always require the LWP to be constant so that the data always need to be stratified by LWP whether Nd or Reff is used for analysis. Can the authors clarify why Nd is more advantageous than Reff for estimating the Twomey effect? Explanations in Section 3.1 are not convincing enough.

*The reviewer is right that the Twomey effect, understood as a radiative effect, has to be considered at constant LWP. This was a sloppy formulation in the Discussion manuscript. What rather was meant, was that Eq. 2 is better formulated with Nd rather than reff: the middle term, $\partial \ln N_{\mathrm{d}} / \partial \ln a$ is much more straightforward evaluated than if one would go for $\partial \ln r_{\mathrm{e}} / \partial \ln a$. In the formulation with Nd, the only other relevant quantity is the vertical wind velocity, while in the formulation with reff one would need to control also for L which is adds a lot of complexity. This clarification is now added to the revised manuscript.*

2. The authors show several lines of evidence that past studies likely underestimated the radiative forcing due to the Twomey effect with some quantitative information of

how large is the underestimates (such as those shown in Figures 1 and 3). I am just wondering if the authors could propose a range of estimate for the radiative forcing that is "corrected" from the existing estimate (like IPCC AR5) accounting for the factors listed in the manuscript that may have caused the underestimate. Such a quantitative estimate would be desirable to show if it is possible.

*The reviewer raises a good point that we internally discussed quite a bit, too. We in the end decided not to provide a new "best estimate". The reason is that although there are a number of studies that address important aspects of the problem and overcome several of the shortcomings listed, none yet does address all. It would this provide the false impression that a solution already exists.*

3. In section 2.1, the authors should explain in more detail why and how the EarthCARE lidar can improve the accuracy of retrieving and discriminating aerosols and clouds, particularly for those of readers who are not familiar with EarthCARE lidar specification. In particular, more explanations would be useful for how ATLID can (i) better distinguish the optically thin clouds and aerosols and (ii) better profile the aerosol extinction, with the capability of HSRL enhanced from CALIOP.

*Two extra sentences explaining this are added to the revised manuscript.*

4. In section 2.2: How can recent geostationary satellites with unprecedentedly high spatial and temporal resolutions provide potentially useful information for horizontal collocation in the context of trajectory approach? For instance, Kikuchi et al. (2018) exploited the high frequency sampling of Himawari-8 to create a new data set of AOD interpolated to the location collocated with clouds that is likely more relevant to CCN.

*This is an excellent point by the reviewer, and the high potential of geostationary satellites increasingly receives attention in the field. A corresponding statement is added.*

[Figure]

5. In section 2.3: Is there any specific way of parameterizing the dry aerosol properties from the humidified one? Some literature information would be desirable to let the readers to have more specific ideas of the issue of swelling.
*Very valid point by the reviewer. We now explicitly explain which parameterisations we think of in getting from humidified to dry aerosol, citing the relevant references.*
* * *

---

## Author Comment (AC2) · 24 Sep 2020

Dear authors, I would like to draw your attention to a recently published ACP paper that makes use of vertically resolved CALIPSO retrievals for investigating co-variability between cloud droplet number concentration (Nd) from MODIS and aerosols (Painemal et al., 2020). We also discuss the advantages of using vertically resolved aerosol properties relative to the common approach of using aerosol optical depth. The material discussed in Painemal et al. (2020) could be relevant to the topic discussed in your manuscript.
*We are really grateful for pointing at this important paper. It is a pity we missed it in the first place, since it already was in Discussions stage during time of writing our review! The paper is now referenced at several instances in the revised manuscript.*

---

## Author Comment (AC3) · 24 Sep 2020

This overview paper is a pretty substantial and concise overview of Twomey effect diagnostics from space, principally with passive solar observations. The paper is generally well-written (save a few passages – something not unexpected given the many co-authors and the unavoidable mixing of styles) and breaks down the problem in an intelligent and intuitive manner. The heart of the paper is eq. (4) which is then further recast as eq. (5). These equations indicate that assessing the strength of the Twomey effect rests on being able to predict the change in cloud droplet number concentration given an anthropogenic CCN perturbation. The latter is not examined; rather the paper focuses on whether the sensitivity of droplet concentration to changes in CCN can be inferred from space observations. The issues investigated are whether aerosols

(and what aerosols in terms of vertical location) can stand-in for CCN and at which level in the cloud the knowledge of the droplet concentration is relevant to calculate the Twomey radiative perturbation. Given the nature of the paper, there is really no original research, but there is plenty of good insight. The paper lacks visual support: there are only three figures in 18 pages. To me at least, it seemed as if the paper loses steam starting in section 4 when text appears to suffer from deteriorating clarity and appears to be more hastily written. But all in all, this is a very noteworthy effort that does not need much of a revision before it becomes a reference to be frequently visited by the aerosol-cloud interaction community.

*We thank the reviewer for their thorough assessment of the manuscript. The impression that sections 4-6 seem to be less substantial is certainly not because these issues are less relevant or that we paid less attention – it is merely the fact that one cannot rely on as large a body of research as is the case for the first two issues (Sections 2 and 3).*

Some remarks/suggested edits:
Line 10 and many instances thereafter: "vertical wind" does not seem the right term; rather people traditionally use the term "updraft velocity", or, given the convention of this paper, "updraught velocity".
*Modified as suggested.*

Line 11: "10s", this read like 10 seconds to me, so better write explicitly "tens".
*Modified as suggested.*

Line 21: "the impossibility" (of retrieving base CCN): Well, some would disagree, and the paper itself does cite Rosenfeld et al. (2016) who claim that such retrieval is possible. See line 289.
*Agreed! The word is changed to "difficulty".*

Line 53: Cloud horizontal extent is actually irrelevant, if the quantity of interest is cloud albedo. Cloud fraction becomes relevant only when the dependences of the Twomey effect on spatial scales is discussed and then only when mixtures of clear and cloudy skies are considered, namely the Twomey effect is expressed in terms of the cloud radiative effect.
*The reviewer is correct, and this mistake is corrected!*

Line 54: "$a_c$ is a monotonic function of $N_d$": only when the cloud condensate is constant.
*The reviewer is right. The statement is corrected by specifying that this is true in the partial-derivative-sense.*

Eq. (2): A derivative of absolute $a_c$ change with respect to a relative (logarithmic) $N_d$ change is shown, while eq. (1) is expressed in terms of relative changes for both quantities. It may make sense to keep these consistent. See also line 81.
*This is a very good suggestion by the reviewer. We opted for modifying Eq. 1 accordingly.*

Line 66: SOLAR zenith angles.
*Modified as suggested.*

Lines 75-79: $N_d$ is also a function of L (you say that actually in line 323), so I don't understand the argument here, which is fundamental for insisting that Twomey effect studies are conducted in terms of $N_d$ (not a directly retrievable quantity) and not $r_e$ (which is directly retrieved). Changes in L can be distributed as both droplet size and droplet number changes, no? See also lines 435-436 about the need to stratify by L when using $r_e$.
*More detail on this is added now. The idea that re and L are both extensive quantities (dependent on mass), Nd is intensive is now explicitly formulated here, too.*

Lines 169-171: Need to clarify that this is the case for passive SWIR observations. Lidar retrievals are discussed elsewhere in the paper.
*The reviewer is right. We added "passive" to the sentence.*

Line 200: I suggest "become less representative of aerosol variability".
*Modified as suggested.*

Line 201: To be consistent with elsewhere in the text: "updraughts".
*Modified as suggested (in fact, ACP encourages British English).*

Lines 271-272: It is implied here that AI is routinely available from space. Is it? For example, MODIS dark target provides AI only over ocean. Is it reliably retrieved? Fig. 2 excludes the land, probably because of this exact unavailability of AI over continents.
*The reviewer has a good point. AI is available, but not very reliable. That information is now added.*

Line 284: The MERRA-2 aerosol re-analysis is also another popular product. Later in lines 287-288, it is not clear how one can evaluate re-analysis aerosol, especially underneath cloud. One has to use observations that are not part of the assimilation process.
*The reviewer is right. A reference to MERRA-2 is added. Indeed, for evaluation one would need other data, such as from the ground; this is clarified now.*

Line 294: I suggest "derivations of supersaturation".
*Modified as suggested.*

P. 12 discussion on $N_d$ retrieval uncertainties: The discussion seem to suggest that higher resolution measurements are needed to reduce cloud heterogeneity effects, yet the retrievals should eventually be coarsened anyway to reduce the random error.
*The reviewer is of course right. The point we wanted to make was probably a bit unclear since we did not provide the precise reference, which is now corrected (Zhang et al., 2016).*

Lines 359 and 362: Deriving cloud base and cloud physical thickness is of course one of the most difficult problems in space-based remote sensing. Lidar can be useful only when the clouds are optically thin (optical thickness below 3-4). So, I wouldn't count too much on space-based lidars for many of the clouds that are relevant to the Twomey effect.
*On the one hand, we agree with the reviewer that this is a difficult problem. On the other hand, a couple of studies are referenced that discuss the problem and propose solutions.*

Line 401: "$\hat{\beta}$ is smaller than unity". Earlier, line 87, it was established that beta is smaller than unity. No range was given for $\bar{\beta}$, but presumably the same implies. Do the authors then mean to say in line 401 that $\hat{\beta}$ is smaller than $\bar{\beta}$?
*It is indeed not completely evident. But what we meant is that it is indeed less than, not equal to, unity (the physically plausible range would include 1). We add the word "somewhat" to make this more clear at this point.*

Line 438: conditions cannot become small, so the authors need to rephrase.
*The reviewer is right. What really was meant is too homogeneous. It is reworded.*

Line 445: I suggest you say "closer to $\sim$50 km scales".
*Modified as suggested.*

Section 6: I found this section about confusing, but I think mostly because of my un-

familiarity with the "regression dilution" concept and the ways its impact is assessed. The term does indeed exist and describes the biasing of the regression slope towards zero values, but you may want to provide a brief definition and description. For people who are familiar with this bias tendency this section may make more sense. Please revisit and ensure that you provide maximum clarity to the uninitiated.
*Accepted, it is indeed helpful to provide some more explanation, which we did in the revision. Also more references are now added.*

Lines 473-474: "the impossibility to retrieve it in cloudy skies". This is a sweeping statement which need some qualifiers. Yes, you can't probably retrieve aerosol under clouds in most situations, but with lidar it is possible both above and below clouds for certain clouds. Also you can retrieve aerosol between individual clouds of a cloud field from both passive and active. Such a cloud field is still "cloudy skies".
*The reviewer is right, this was a sloppy formulation. We revise to say "below clouds".*

Line 480: I suggest "in addition to retrievals".
*This was confusing indeed, but meant in a slightly different way. Reworded to "The hygroscopic swelling can be addressed by parameterisations that use retrievals and ancillary data to compute the swelling."*

Line 486: I suggest "relates imperfectly to the $N_d$".
*Modified as suggested.*

Line 487: You mean sensitivities less than one? I don't understand as it is currently written.
*Indeed, the formulation the reviewer suggests is better!*

Line 504: I suggest "quantification supported by data"
*Modified as suggested.*